

**Insights into the high temporal variability of atmospheric carbon dioxide**
**($CO_2$) at a suburban station in the Indo-Gangetic Plain**
Vimal Jose Vazhathara[1, *], Ravi Kumar Kunchala[1], Sajeev Philip[1], Jaswant Rathore[1], Dilip
Ganguly[1], Sagnik Dey[1,2], Yutaka Matsumi[3,4] and Prabir K. Patra[3,5]
[1]Centre for Atmospheric Sciences, Indian Institute of Technology Delhi, New Delhi, India.
[2]Centre of Excellence for Research on Clean Air, Indian Institute of Technology Delhi, New
Delhi, India
[3]Research Institute for Humanity and Nature, Kyoto, Japan
[4]Institute for Space-Earth Environmental Research, Nagoya University, Nagoya, Japan
[5]Japan Agency for Marine-Earth Science and Technology (JAMSTEC), Yokohama, Japan.
*Correspondence to: Vimal Jose Vazhathara (vimaljosevazhathara@gmail.com), Centre for
Atmospheric Sciences, Indian Institute of Technology Delhi, New Delhi, 110016, India
**Abstract**
The unusual weather patterns and large anthropogenic emissions over the Indo-Gangetic Plain
(IGP) make it a significant hotspot of greenhouse gases like carbon dioxide ($CO_2$). Given the
paramount significance of the IGP, a GHG observatory was set up at a suburban monitoring
station, Sonipat, Haryana (28.95 °N, 77.10 °E), in the Delhi National Capital Region. Using
continuous measurements of $CO_2$ using a laser-based cavity ring-down spectroscopy (CRDS)
technique, we investigated the temporal evolution of $CO_2$ mole fraction from February 2023 to
January 2025. We observed an annual average $CO_2$ mole fraction of 440.8 ± 19.7 parts per
million (ppm) with an unusually strong seasonal variability ranging from 422.6 ± 23.3 to 456.4
± 30.8 ppm in monsoon and post-monsoon, respectively. A strong $CO_2$ diurnal amplitude of
29 ppm in May and 63 ppm in October was observed mainly due to seasonal changes in
boundary layer mixing and biospheric activity. Further investigation of the drivers of this
unique feature (strong seasonal and diurnal $CO_2$ variability) over IGP revealed a strong contrast





to other global monitoring stations in the same latitude band. A strong correlation between $CO_2$
and $CH_4$ indicated a co-located emission source, while the strong positive correlation between
$CO_2$ and carbon monoxide (CO) during post-monsoon indicates the footprint of crop residue
burning on $CO_2$ mole fraction. We demonstrate that the high temporal $CO_2$ variability in the
IGP is driven by the interplay of local anthropogenic and biomass burning emissions,
biospheric fluxes, and prevailing meteorology.

## 1. Introduction

Carbon dioxide ($CO_2$) is the major greenhouse gas (GHG) contributing to climate change and
global warming ( IPCC, 2021; Fawzy et al., 2020). Due to the long lifetime and high radiative
forcing potential, $CO_2$ can have a significant impact on global and regional climate (Wang et
al., 2010). The atmospheric mole fraction of $CO_2$ has increased from 278 parts per million
(ppm) in the pre-industrial period to 427 ppm in 2025 (NOAA, https://gml.noaa.gov). This
rapid increase in the atmospheric fraction of $CO_2$ is primarily due to the combustion of fossil
fuels, cement manufacture, deforestation, and other industrial processes ( Stocker et al., 2013;
Huang et al., 2016; Yoro and Daramola, 2020). A comprehensive understanding of the sources
and sinks of $CO_2$ is critical for developing national policies to mitigate climate change impacts.

India is the third highest $CO_2$ emitting nation (8% of total global $CO_2$) in the last decade

as reported by the Global Carbon Project (GCP) (Friedlingstein et al., 2025; Le Quéré et al.,
2018). In particular, the Indo-Gangetic Plain (IGP) region is one of the hotspots for atmospheric
$CO_2$ mole fraction primarily due to the large fossil fuel emissions and adverse meteorology
(Kuttippurath et al., 2022; Singh et al., 2022). Over the past few decades, the IGP region has
witnessed rapid urbanisation, industrialisation, and agricultural intensification, leading to
significant changes in land-use patterns and GHG emissions (Yoro and Daramola, 2020).
Mitigation of anthropogenic $CO_2$ emissions over the highly populated IGP region is crucial for
reducing high atmospheric $CO_2$ mole fraction build-up. Gaining a better understanding of the
magnitude of $CO_2$ sources and sinks and the local drivers of $CO_2$ temporal variability over the
IGP region is therefore important.

The continuous monitoring of ground-based $CO_2$ is of utmost importance for the

inverse modelling approaches to understand the sources and sinks of $CO_2$. Although GHG mole
fraction have been monitored over various parts of the globe for decades, monitoring stations
of GHGs in India are limited (Chakraborty et al., 2020; Kumar et al., 2021; Patra et al., 2013;





Tiwari et al., 2013) The Cape Rama (15.08° N, 73.83° E), situated on India's southwest coast, was the first Indian monitoring station tracking $CO_2$ mole fraction from 1993 to 2002 (Bhattacharya et al., 2009; Patra et al., 2011; Rayner et al., 2008). Recently, several monitoring stations have been established over different parts of India to measure the GHGs (Chandra et al., 2016; Jain et al., 2021; Mahesh et al., 2015; Metya et al., 2021; Nomura et al., 2021; Pathakoti et al., 2023; Sreenivas et al., 2016; Thilakan et al., 2023; Tiwari et al., 2014). Some aircraft-based (Niwa et al., 2012; Patra et al., 2011; Schuck et al., 2012; Zhang et al., 2007) and satellite-based (Das et al., 2023; Kunchala et al., 2022; Nalini et al., 2019; Philip et al., 2022; Xiong et al., 2009) studies have also been conducted in the past. The incorporation of the regional in situ and aircraft-based measurements along with satellite column $CO_2$ retrievals reduced uncertainties in top-down $CO_2$ flux estimations (Huang et al., 2008; Niwa et al., 2012; Zhang et al., 2014). These studies highlighted the importance of regional ground-based observations in constraining Indian carbon cycle dynamics. However, the IGP region still lacks continuous measurements to track temporal evolution of atmospheric $CO_2$ mole fraction except for one station in Mohali (Thilakan et al., 2023).

To comprehensively understand the temporal $CO_2$ variability along with its magnitude and the drivers of $CO_2$ in the IGP region, we have conducted atmospheric $CO_2$ mole fraction measurements at Sonipat, a suburban station in the IGP region upwind of Delhi. The continuous measurements from February 2023 to January 2025 were conducted using the laser-based cavity ring-down spectroscopy technique. Here, we investigate the novel characteristics of the seasonal and diurnal variability of atmospheric $CO_2$ mole fraction over the Sonipat monitoring station. We then identify the key drivers of the observed temporal $CO_2$ variability over the Sonipat station to gain insights into the carbon cycle dynamics of the entire IGP region.

## 2. Materials and methods

### 2.1 Monitoring station

The measurements in this study were carried out at the Indian Institute of Technology Delhi (IIT Delhi) Centre for Atmospheric Sciences (CAS) - Atmospheric Observatory situated at Sonipat campus (28.95° N, 77.10° E, 228 m amsl altitude). Sonipat is an upwind suburban region of Delhi situated in the north Indian state of Haryana and a part of the Delhi National Capital Region (NCR). The monitoring station is surrounded by agricultural fields, a National Highway, and academic institutions. Figure 1 shows the location map of the monitoring station. The climatic conditions over this site are similar to Delhi which has sweltering summers, damp or moist monsoons (June - September), and extreme winters. Similar to Delhi, this region also





has frequent haze and smog with low visibility during winter (December - February) and post-
monsoon (October - November) seasons. During post-monsoon season, Sonipat station
experiences large transport of pollutants from the North-West direction. In addition to the
pollutant transport, several local emissions sources exist in the region, such as small industries,
vehicular sources, and local biomass burning.

**2.2 Local measurements**


**2.2.1 GHG measurements**


This study utilised the PICARRO G2301 GHG analyzer to measure major atmospheric GHG
mole fraction. The PICARRO analyzer employs the Cavity Ring-Down Spectroscopy (CRDS)
technique at 0.5 Hz to measure $CO_2$ mole fraction. The CRDS technique utilises the ring-down
time of light intensity within the cavity to determine the mole fraction of $CO_2$, a method
fundamentally different that other measurement techniques such as other techniques such as
Non-dispersive Infrared Spectroscopy (NDIR) and Fourier Transform Infrared Spectroscopy
(FTIR). The long sample interaction path length (approximately 20 km) is a characteristic of
CRDS, which enhances sensitivity compared to conventional techniques based on light-
intensity absorption. The cavity pressure operates at a very low pressure of 140 Torr. This
isolates a single spectral feature with a resolution of 0.0003 $cm^{-1}$, ensuring a linear relationship
between peak height or area and mole fraction. The CRDS offers precise and highly sensitive
measurements of gases in ambient air with a temporal resolution of 5 seconds. The technique
has been well validated for the measurements of atmospheric CO, $CO_2$, and $CH_4$ mole fraction,
globally and over some Indian monitoring stations (Chandra et al., 2016; Chen et al., 2013;
Jain et al., 2021).

In this study, the cavity temperature was maintained at 45°C throughout the

measurement period to ensure the necessary etalon mechanical stability of the measurement
cavity. The sample air was taken from the top of the building and above the tree canopy (10
meters above the instrument housing) using an external vacuum pump and Teflon tube at ~400
SCCM flow rate.

To better interpret the temporal variability in the atmospheric $CO_2$ mole fraction, we

calculated the background $CO_2$ mole fraction at Sonipat. The background mole fraction are
typically calculated from measurements over pristine sites free of local emission sources. The
Sonipat station, lying on the upwind side of Delhi, is a suburban station with relatively cleaner
air when compared to the urban city centre. However, Sonipat cannot be considered a pristine
site due to the impact of local emissions from nearby industries and national highways.



Typically, two techniques are used to calculate background $CO_2$ mole fraction at such
monitoring stations. The fifth percentile method is based on the fifth percentile of the daily data
to calculate the background mole fraction (Ammoura et al., 2014; Chandra et al., 2016; Jain et
al., 2021). The adaptive diurnal minimum variation selection (ADVS) method considers the
diurnal minimum value as the daily background value (Apadula et al., 2019; Yuan et al., 2018).
In this study, the comparison between the fifth percentile and the ADVS methods showed
similar $CO_2$ background values (see Fig. S1). Therefore, we adopted one of the methods
(ADVS) here to report the background $CO_2$ mole fraction at Sonipat station. The excess $CO_2$
mole fraction were then estimated by subtracting the hourly averaged values of $CO_2$ from the
background mole fraction.

The measurements of the atmospheric $CH_4$ mole fraction were also conducted with the

PICARRO G2301 GHG analyser. The GHG analyser employs the CRDS at 0.5 Hz to measure
$CH_4$ mole fraction. The mole fraction of $CH_4$ were determined using the ring-down time of
light intensity, similar to $CO_2$ mole fraction.   Calibration was performed following the
guidelines of the National Oceanic and Atmospheric Administration Earth System Research
Laboratories (NOAA-ESRL, 2020) and the Integrated Carbon Observation System (ICOS)
protocol (Laurent, 2016), using NOAA standard calibration cylinders. Further details of the
calibration process are provided in Supplementary Section S1.

**2.2.2 Trace gas measurements**
In addition to the measurements of $CO_2$ and $CH_4$, we also utilised the measurements of trace
gases to establish the species interrelationships and to identify drivers of GHG sources. We
used a compact air quality measurement instrument with gas sensors (CUPI-G) to collect
continuous measurements of air pollutants, including fine particulate matter ($PM_{2.5}$), nitric
oxide (NO), nitrogen dioxide ($NO_2$), and carbon monoxide (CO). The sensors used in CUPI-G
are a palm-sized optical $PM_{2.5}$ sensor developed by Panasonic, the CO-B4 Carbon Monoxide
Sensor, and the NO-B4 Nitric Oxide Sensor, respectively. The CUPI-G was deployed on the
roof of the I-Techpark building at the Sonipat campus of IIT Delhi.

**2.2.3 Local meteorology measurements**
A Vaisala Ceilometer lidar CL61 was installed on the rooftop of the I-Techpark building at IIT
Delhi's Sonipat campus at the same location as the GHG analyser is located. The CL61 system
is designed to provide real-time measurements of cloud base height (CBH) for up to five layers,
along with depolarisation measurements, under all weather conditions. To determine the



Planetary boundary layer height (PBLH) from the range-corrected attenuated backscatter data,
the gradient method (Summa et al., 2013) and the Wavelet Covariance Transform (WCT)
method (Baars et al., 2008) were employed. Further details on PBLH calculations can be found
in (Rathore et al., 2025). An automatic weather station (AWS) by Geonica, installed on the I-
Tech building rooftop, collected meteorological data at 5-minute intervals. The data, including
ambient temperature, relative humidity (RH), atmospheric pressure, wind speed and direction,
precipitation, and incoming solar radiation, was retrieved using Datagraph-W4K 2.1.3.0
software and exported in CSV format. All sensors were meticulously calibrated and regularly
cleaned to ensure accuracy and reliability.

**2.3 Auxiliary data**
**2.3.1 ObsPack Data**
To compare the seasonality of atmospheric $CO_2$ of Sonipat with other non-Indian sites in the
same latitudinal band, we used the obspack_co2_1_GLOBALVIEWplus_v10.1_2024-11-13
(Schuldt et al., 2024). This dataset is constructed using the Observation Package (ObsPack)
framework (Masarie et al., 2014). This product includes 625 atmospheric carbon dioxide
datasets from observations made by 79 laboratories from 28 countries. The ObsPack dataset
provides data for the period 1957-2023. We used the five-year averaged data for all sites except
one (Boulder Atmospheric Observatory, Colorado) for 2018-2022 to further compare the
seasonality over different locations across the globe.

**2.3.2 Satellite $CO_2$ retrievals**
Along with the ground-based in situ $CO_2$ measurements at the Sonipat monitoring station, we
also used column average dry air $CO_2$ mole fraction ($XCO_2$) retrievals from the Orbiting
Carbon Observatory-2 and 3 satellites (OCO-2 and OCO-3). The OCO-2 satellite provides data
at a temporal frequency of 16 days with a spatial resolution of 1.29 km × 2.25 km (for nadir
observations) (Crisp et al., 2017; Eldering et al., 2017). We used the bias-corrected OCO-2
v11.1r data product for the period from February 2023 to December 2024. The OCO-3 satellite
provides $XCO_2$ data at a temporal frequency of 16 days with a spatial resolution of 1.60 km ×
2.25 km (nadir observation) which increases the swath area from ~3.0 km$^2$ to ~3.5 km$^2$. We
used bias-corrected OCO-3 v10.4r data product (Eldering et al., 2019; Srivastava et al., 2020)
for February 2023 to December 2024.

**2.3.3 FluxSat GPP**





To study the Gross Primary Production (GPP) fluxes over Sonipat, we used FluxSat v2.2 native
GPP product computed at the spatio-temporal resolution of the MCD43C data set (daily at
0.05° spatial resolution (Schaaf et al., 2002; Wang et al., 2018). FluxSat v2.2 has been derived
from the MODerate resolution Imaging Spectroradiometer (MODIS) instruments on the NASA
Terra and Aqua satellites using the collection 6.1 MCD43C Bidirectional Reflectance
Distribution Function (BRDF)-Adjusted Reflectances (NBAR) (Joiner et al., 2018; Joiner and
Yoshida, 2020; Schaaf and Wang, 2021). FluxSat v2.2 is "calibrated" using a set of the
FLUXNET 2015 and OneFlux tier 1 (publicly released) eddy covariance (EC) data and has
been compared with independent data (i.e., not used in the calibration) as validation. We used
Global Gross Primary Production (GPP) estimates for 2023 in this study.

**2.3.4 Ecosystem-proxy variables**
We used two key ecosystem-proxy variables to study the carbon cycle dynamics of the Sonipat
station and the IGP region. The Normalised Difference Vegetation Index (NDVI) version 5
data from the Advanced Very High-Resolution Radiometer (AVHRR) was used here (Vermote
and NOAA CDR Program, 2018). This dataset consists of daily NDVI values retrieved from
the National Oceanic and Atmospheric Administration's (NOAA) Climate Data Record (CDR)
of AVHRR Surface Reflectance. The NDVI CDR summarises the surface vegetation coverage
activity based on measurements in the red and near-infrared spectral bands. The NDVI CDR
provides daily output on a global grid with a resolution of 0.05 degrees latitude by 0.05 degrees
longitude from 1981 to the present.

To understand the photosynthetic capacity of the regional ecosystem to assimilate

atmospheric $CO_2$, we used the Solar Induced chlorophyll Fluorescence (SIF) retrievals from
the OCO-2 satellite (Frankenberg et al., 2014). The OCO-2 provides SIF data at a temporal
resolution of 16 days and a spatial resolution of 1.35 km × 2.25 km. The estimation of SIF
relies on evaluating the in-filling of solar Fraunhofer lines at 757 nm and 770.1 nm surrounding
the $O_2$ A-band (Frankenberg et al., 2014; Sun et al., 2018). We used bias-corrected SIF data
from OCO-2 v11r and v11.2r SIF data products for February 2023 to December 2024.

**2.4 Models**
**2.4.1 JAMSTEC's MIROC version 4 atmospheric chemistry-transport model (MIROC4-**
**ACTM)**
We used the Model for Interdisciplinary Research on Climate version 4 ( MIROC4;Watanabe
et al., 2008), an atmospheric general circulation model (AGCM)-based chemistry-transport



model (MIROC4-ACTM; Patra et al., 2018), to simulate $CO_2$ mole fraction for this study.
Simulations of long-lived gases ($CO_2$, $CH_4$, $N_2O$, $SF_6$) were performed at a horizontal
resolution of T42 spectral truncations (~2.8° latitude–longitude grid) with 67 vertical hybrid-
pressure layers between the Earth's surface and 0.0128 hPa (~80 km) (Bisht et al., 2021;
Chandra et al., 2021; Patra et al., 2017, 2018). $CO_2$ tracers were simulated corresponding to the
fossil fuel combustion ($FFCO_2$), land biosphere fluxes ($LBCO_2$), fire emissions ($CO_{2fire}$), and
ocean exchanges ($CO_{2ocn}$) from different sets of prior (bottom-up) emissions (Chandra et al.,
2022). $FFCO_2$ was simulated using the gridded fossil fuel emission dataset ( GridFED; Jones
et al., 2021). $LBCO_2$ tracers were simulated using two sets of terrestrial biosphere fluxes from
the Carnegie-Ames-Stanford Approach (CASA) biogeochemical model (Randerson et al.,
1997) and Vegetation Integrative Simulator for Trace Gases (VISIT) (Ito, 2019). For this study,
we use simulated total $CO_2$ mole fraction and tracers.
**2.4.2 CarbonTracker (CT) inverse model**
To understand the temporal pattern of atmospheric $CO_2$ mole fraction over the study station
and the IGP region, we used simulated $CO_2$ mole fraction from an inverse modelling
framework CarbonTracker (CT) (Peters et al., 2005). Here, we used the CarbonTracker 2022
release (CT2022) which incorporated two-way nesting of the offline atmospheric tracer
transport model TM5 supporting coarse-resolution data globally and high-resolution data
regionally (Krol et al., 2004). The TM5 model in CT2022 was driven with meteorology from
the ERA-interim reanalysis provided by the European Center for Medium-Range Weather
Forecasts (ECMRWF) {Citation}. The CT2022 inverse model simulated atmospheric $CO_2$
mole fraction by correcting the prior specifications of $CO_2$ sources and sinks in the model by
assimilating global in situ observations. In this study, we used CT2022-simulated $CO_2$ mole
fraction from February 2023 to October 2023.
**2.4.3 GEOS-Chem inverse model**
To study the seasonality of the fluxes over Sonipat, we used a four-dimensional variational
(4D-Var) assimilation system with the GEOS-Chem global chemical transport model
(CTM;Philip et al., 2019, 2022). The GEOS-Chem 4D-Var system was constrained with $XCO_2$
retrievals from the OCO-2 satellite (Philip et al., 2022), following the protocol of the OCO-2
v10 Multi-model Intercomparison Project (MIP) (Byrne et al., 2017; Liu et al., 2014). The Net



Ecosystem Exchange (NEE) fluxes for 2023 at a spatial resolution of 1° × 1°, constrained with
the OCO-2 Land Nadir and Land Glint observational modes are used here.

**266    2.4.4 Mi CASA terrestrial biospheric model**

We simulated $CO_2$ fluxes from a terrestrial biospheric model (TBM) have also been used in
this study. The Más informada Carnegie-Ames-Stanford-Approach (Mi CASA) model (Weir,
2024), a comprehensive update to the CASA – Global Fire Emissions Database, version 3
(CASA-GFED3) product, was utilised here (Chen et al., 2023; Potter et al., 1993). Mi CASA
provides daily global data at 0.1° resolution from January 2001 to December 2023. This
includes carbon flux variables from sources such as net primary production (NPP),
heterotrophic respiration (Rh), wildfire emissions (FIRE), and fuel wood burning emissions
(FUEL). The model is driven with meteorological data from NASA's Modern-Era
Retrospective analysis for Research and Application, Version 2 (MERRA-2). Previous studies
used the MERRA-driven CASA GFED to investigate the carbon cycle dynamics (Campbell et
al., 2008; Hammerling et al., 2012; Kawa et al., 2010; Ott et al., 2015; Weir et al., 2021a, b).
We used Mi CASA model simulated NEE, NPP, and Rh fluxes over the Sonipat station for this
study.



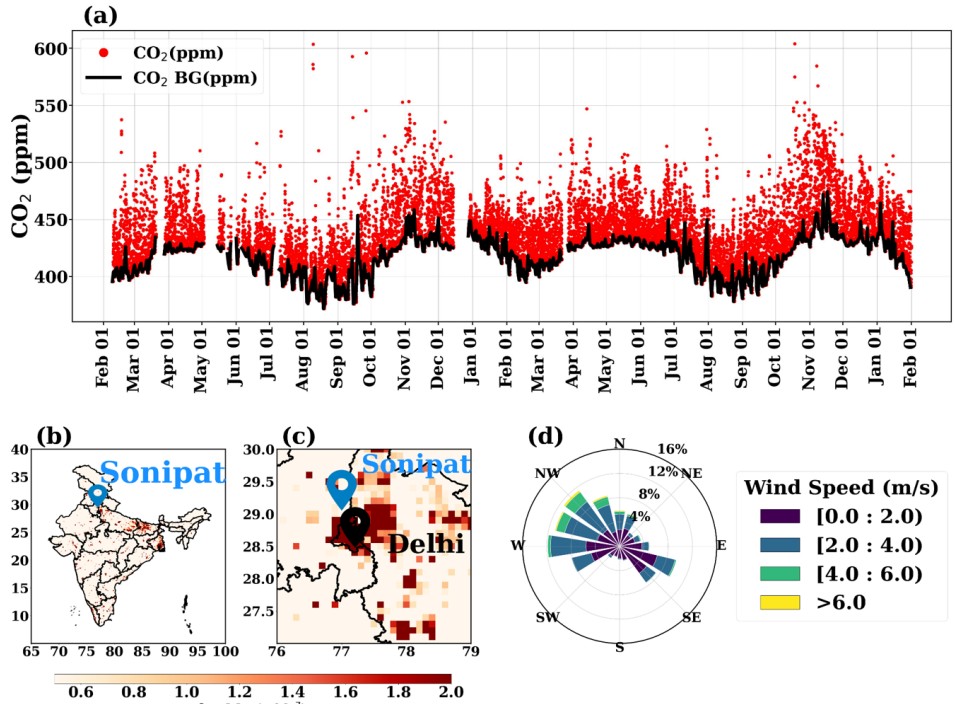


**Figure 1:** (a) Hourly averaged time series of atmospheric $CO_2$ (red) mole fraction for the study period (February 2023 to January 2025) over Sonipat. The thick black line represents the background mole fraction estimated using the Adaptive Diurnal least Variation Selection (ADVS). Anthropogenic $CO_2$ emissions over (b) India and (c) Sonipat/Delhi are derived from the EDGAR emission inventory for 2021. (d) Annually averaged wind patterns over Sonipat for February 2023 – January 2024.

## 3. Results and discussions

### 3.1 $CO_2$ measurements at Sonipat station

Figure 1(a) shows hourly averaged time series of atmospheric $CO_2$ mole fraction at the Sonipat station from February 2023 to January 2025. During this period, hourly $CO_2$ varies in the range from ~380 ppm to ~ 550 ppm, with the highest values observed in November 2024 indicating the large variability in the regional $CO_2$ build-up at the study location. The strong seasonal and diurnal variations are evident during the entire study period. In general, minimum variability and lowest mole fraction of $CO_2$ are found from July to August, while strong variability and





high mole fraction are visible from October to November. We found annual mean $CO_2$ mole
fraction of 440.8 ± 19.7 ppm during the study period. Table S1 compares the annual mean $CO_2$
mole fraction values with other measurement network stations in India. Interestingly, the
annual mean values for different stations in India, even rural sites like Gadanki and urban sites
like Ahmedabad, have consistent values.

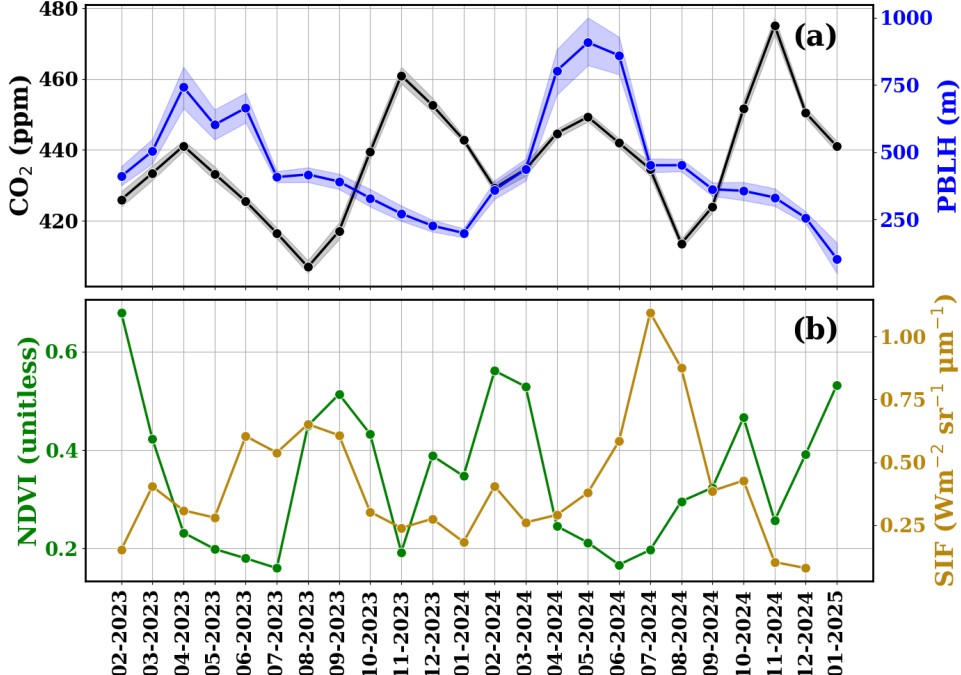


**Figure 2:** (a) Monthly variations of atmospheric $CO_2$ mole fraction (black) and PBLH (blue) and (b) NDVI (green) and SIF (olive green) over the Sonipat monitoring station during the study period.

Figures 1(b) and 1(c) illustrate the annual mean anthropogenic $CO_2$ emissions over
India and Sonipat for 2021 based on the EDGAR emission inventory, and it is observed that
the Delhi NCR is a hotspot of anthropogenic $CO_2$ emissions. Figure 1(d) shows that the
dominant wind direction over Sonipat during the study period was from the northwest,
indicating significant influence from upwind sources of pollution and greenhouse gases. To
better understand the local meteorology, we analysed the seasonal variations of various
meteorological parameters such as air temperature, wind speed, wind direction and relative
humidity (Fig. S2). Figure S3 presents the seasonally averaged wind rose diagrams over



Sonipat. It is evident that the predominant wind pattern is from the northwest. In this study, we
focus on seasonal and diurnal $CO_2$ variability and compare these patterns with other stations in
India and the same latitudinal band across the globe to uncover the unique aspects of $CO_2$
dynamics over Sonipat and the IGP as well.

**3.2 Seasonal variability**

**3.2.1 Seasonality of in situ observations**

Figure 2 shows the monthly mean atmospheric $CO_2$ mole fraction during the study period. The
shaded region represents a 95 percent confidence interval of the monthly mean $CO_2$ mole
fraction.  The monthly mean mole fraction of $CO_2$ shows a maximum in November (post-
monsoon season) and a minimum in August (monsoon season) during both years.  The average
seasonal mean values of $CO_2$ observed during different seasons are 440.8 ± 19.7 ppm (pre-
monsoon), 422.6 ± 23.3 ppm (monsoon), 456.4 ± 30.8 ppm (post-monsoon), and 440.5 ± 19.7
ppm (winter).
The seasonal cycle of $CO_2$ is mostly governed by the strength of emission sources,
photosynthetic activity (biospheric fluxes), local meteorology and atmospheric transport. One
of the key factors that controls the local variability of atmospheric $CO_2$ is planetary boundary
layer height (PBLH). This is the lowest layer within the troposphere, where temperature and
wind speed variations are integral in modulating its height. Strong vertical mixing in a well-
developed boundary layer can dilute GHG mole fraction near the surface. Therefore, the
seasonal changes in the PBLH affect the atmospheric $CO_2$ mole fraction near the surface.
Figure 2(a) shows that the minimum values in PBLH during pre-monsoon months and
maximum values during winter. During pre-monsoon, deep convection due to the well-
developed PBLH from the surface to the upper troposphere results in lower mole fraction as
compared to the winter months (Baker et al., 2012; Kar et al., 2004; Park et al., 2009; Patra et
al., 2011; Randel and Park, 2006).
To better understand the seasonal patterns of $CO_2$ mole fraction, we examined its
relationship with the normalised difference vegetation index (NDVI) and solar-induced
fluorescence (SIF). Both NDVI and SIF are widely used indicators of vegetation cover and
photosynthetic activity (Aburas et al., 2015; Nath, 2014). Our analysis shows a strong inverse
relationship between $CO_2$ levels and NDVI, as illustrated in Fig. 2. Figure 2(b) reveals that
vegetation growth starts with the onset of the monsoon season. An enhanced vegetation cover





over the region from August and a noticeable decrease in atmospheric $CO_2$ mole fraction is
evident. Increased vegetation cover shows an increase in the photosynthetic carbon uptake by
the biosphere, which decreases atmospheric $CO_2$ mole fraction. However, as vegetation activity
declines from the post-monsoon to winter and pre-monsoon seasons, photosynthetic carbon
uptake decreases, leading to a rise in atmospheric $CO_2$. The negative correlation of NDVI
versus $CO_2$ mole fraction was found for most locations in India (Metya et al., 2021; Sreenivas
et al., 2016; Tiwari et al., 2014), indicating the strong dependence of $CO_2$ seasonality on the
local vegetative carbon uptake.

A sharp decrease in seasonal mean (~ 18 ppm) is visible from pre-monsoon to monsoon.

This is attributed to the enhanced photosynthetic activity around the measurement site due to
the availability of large soil moisture. A further decrease in $CO_2$ mole fraction is also observed
as the monsoon progresses, with minimum mole fraction observed in August. The decreases in
temperature (due to cloudy and overcast conditions prevailing during these months) reduce leaf
and soil respiration, which contributes to the enhancement of carbon uptake (Jing et al., 2010;
Patil et al., 2014). Further, an increase in $CO_2$ mole fraction (~ 34 ppm) is observed during
post-monsoon, which is associated with higher ecosystem productivity (Sharma et al., 2014)
and an enhancement in soil microbial activity (Kirschke et al., 2013). The gradual decrease in
NDVI during this period indicates a decrease in $CO_2$ uptake by vegetation. This season
coincides with crop-burning episodes in northern India. The crop-burning residue activities
significantly contribute to the increase in $CO_2$ mole fraction. A sharp decrease (~ 16 ppm) in
seasonal mean during the winter is evident compared to post-monsoon. The shallow PBLH and
winds from western IGP that transport crop-burning residue contribute to the enhanced mole
fraction during winter. Table S2 compares the seasonal amplitude and the peak and draw-down
months over the measurement site with similar studies over India. Sonipat exhibits higher
seasonal amplitudes than other sites. However, a similar pattern in $CO_2$ peak and drawdown
months is evident in other monitoring stations.
**3.2.2 Constraints from model and satellites**
Figure 3 shows the comparison of ground-based mole fraction of $CO_2$ with the CarbonTracker
inverse model (CT2022) simulated mole fraction (see different y-axis). The model outputs
beyond October 2023 were not publicly available. In general, the CT2022 model-simulated
mole fraction are much lower than observed mole fraction at the Sonipat station. The
discrepancy could be mainly due to the representativeness issue, due to the coarser model



spatial resolution. Nevertheless, the seasonal pattern of $CO_2$ mole fraction simulated with the

model is in broad agreement with observations (Fig. 3). The CT2022 model simulates the

seasonal variability with a minimum mole fraction of 416 ppm during September, whereas in

situ measurements show a minimum mole fraction of 407 ppm during August. The CT2022

model exhibits higher mole fraction during the pre-monsoon season, similar to in situ data.

Note that most global and regional chemical transport models were unable to reproduce the

large seasonal magnitude of surface-based measured atmospheric $CO_2$ mole fraction for any of

the monitoring stations in India with different land ecosystems (Lin et al., 2018; Philip et al.,

2022).

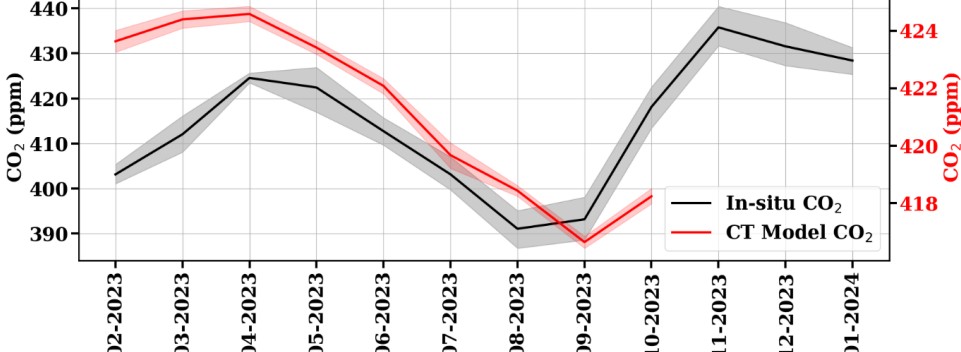

**Figure 3:** Monthly mean background $CO_2$ mole fraction over Sonipat (estimated using ADVS method) compared to CarbonTracker (CT2022) model simulated values at daytime (13:00 – 16:00). Note that the left y-axis represents surface mole fraction from in situ measurements, and the right y-axis represents CT2022-simulated mole fraction.



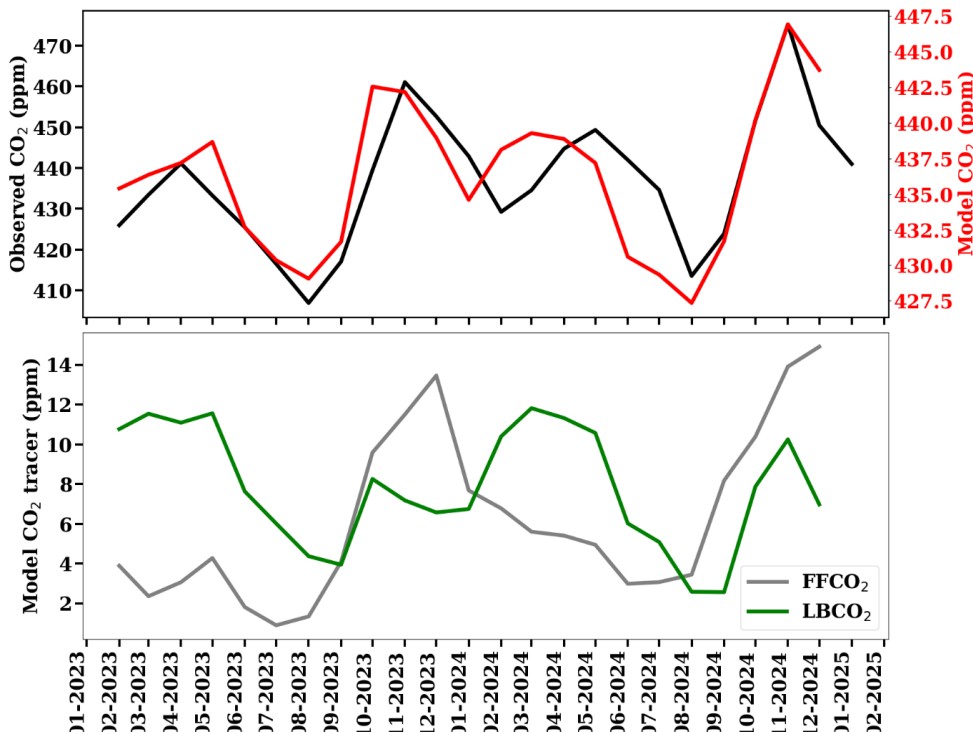


**Figure 4:** (a) Comparison of simulated mole fraction of atmospheric $CO_2$ from MIROC-
ACTM with in situ measurements at Sonipat and (b) monthly averaged time series of different
tracers from the MIROC-ACTM.


Figure 4(a) presents the comparison of atmospheric mole fraction of $CO_2$ over Sonipat
with simulated mole fraction of $CO_2$ from the MIROC4-ACTM model. The model has well
captured the seasonal pattern of $CO_2$, but it fails to capture the seasonal amplitude over Sonipat.
The highs during post-monsoon and the drawdown during pre-monsoon show a strong
correlation with in situ measurements. Figure 4(b) presents the monthly averaged time series
of model-simulated $CO_2$ tracers. The fossil fuel tracer (FFCO$_2$) exhibits a peak in post-
monsoon with a gradual decrease in winter and a drawdown in monsoon, which coincides with
observed $CO_2$ mole fraction. The post-monsoon peak can be attributed to the added emissions
from crop residue burning, which is a characteristic of the site. The drawdown in monsoon can
be attributed to the added soil moisture and increased $CO_2$ uptake by plants during this time
(further discussed in section 3.4). LBCO$_2$ presents a peak during pre-monsoon and a drawdown
in monsoon. The peak can be attributed to dry soil conditions and a lack of vegetation during





this time. A similar enhancement of LBCO$_2$ is visible during the post-monsoon season, which
coincides with the harvest period over the site. The lack of vegetation with added CO$_2$ from
crop residue burning contributes to this enhancement. The drawdown in mole fraction during
the monsoon season is due to wetter soil conditions and enhanced biospheric activity.

416   Figure 5 compares XCO$_2$ from OCO-2 and OCO-3 satellites with ground-based CO$_2$

measurements at Sonipat during the study period. The XCO$_2$ reveals a similar seasonal pattern
with high mole fraction during the pre-monsoon season, followed by a dip in mole fraction
during the monsoon season and a further gradual increase in mole fraction during the post-
monsoon and winter months. Although the satellite column data captures the monthly
variability reasonably well, it fails to capture the sharp increase in mole fraction during the
post-monsoon. This enhancement during the post-monsoon season can be attributed to crop
residue burning over the monitoring station and the added transport from Punjab (see Section
3.5). This highlights the inability of high-resolution satellite data to capture enhancements from
local sources.

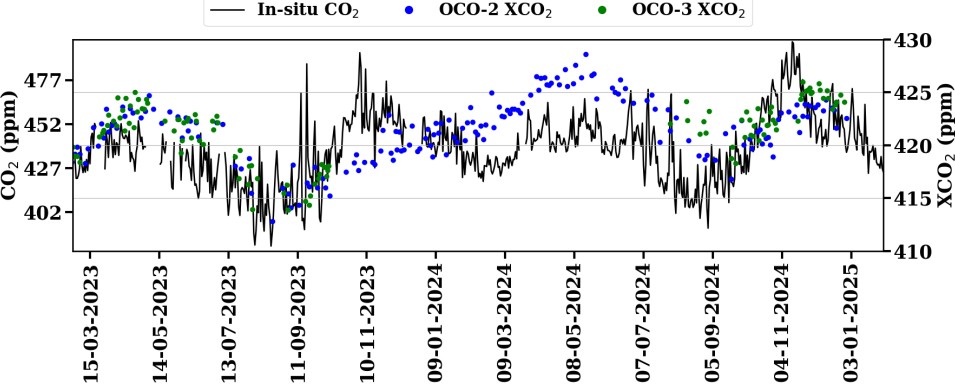



**Figure 5:** Daily variations of atmospheric CO$_2$ mole fraction from in situ measurements over
Sonipat (left y-axis) with column average CO$_2$ mole fraction (XCO$_2$) from the OCO-2 (ppm)
and OCO-3 (ppm) satellite instruments (right y-axis).

**3.2.3 Comparison with data from other monitoring stations**



Figure 6(a) presents the monthly averaged variation of $CO_2$ over Sonipat during the study
period (SNT) with other measurement sites in the same latitudinal band (5° N – 40° N). The
other sites used for comparison include five Indian monitoring stations and six international
stations. The five Indian sites that reported $CO_2$ mole fraction were Shadnagar (SDN; 17.09° N,
78.2° E), Gadanki (GDN; 13.50° N, 79.20° E), Mohali (MHL; 30.67° N, 76.73° E), Sinhagad
(SNG; 18.21° N, 73.45° E) and Ahmedabad (AHM; 23.03° N, 72.55° E). Apart from Indian
sites, six sites in the same latitudinal band (using ObsPACK) were compared. These sites were
Mauna Lou (MLO; 19.54° N, 155.58° W), South Carolina (SCT; 33.40° N, 81.83° W),
Shenandoah National Park (SNP; 38.61° N, 78.35° W), Walnut Grove, (WGC; 38.26° N,
121.49° W), Moody (WKT; 31.31° N, 97.33° W) and Boulder (BAO; 40.05° N, 105.00° W).
Figure 6b compares the seasonal amplitude of the sites in the chosen latitudinal band. The inset
in Fig. 6b shows the location of all the measurement sites. For all non-Indian sites except BAO,
the five-year average (2018-2022) has been chosen for the seasonality. For BAO, 2011-2016
has been used for this study.





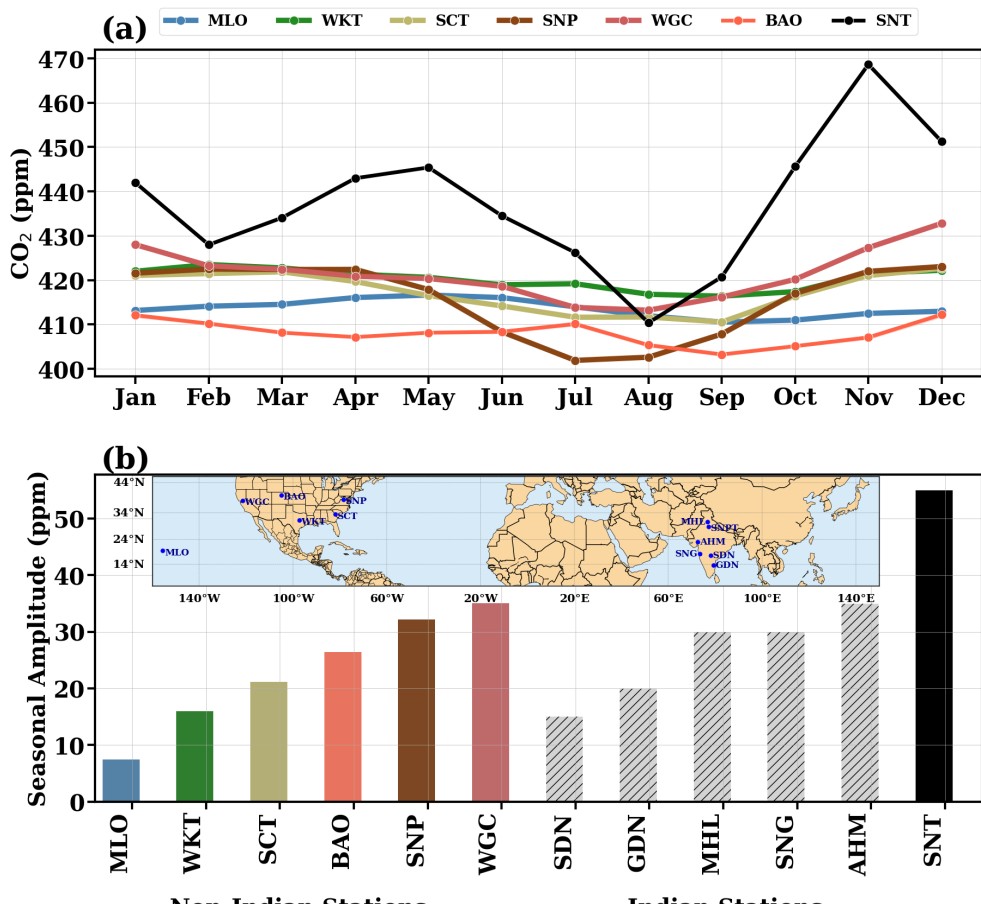

**Figure 6:** (a) Comparison of the seasonal variability of atmospheric $CO_2$ over Sonipat monitoring station with various locations in the same latitudinal band. (b) Comparison of the seasonal amplitude between Indian (coloured bars) and international monitoring stations (grey bars). Indian stations include Shadnagar (SDN), Sinhagad (SNG), Ahmedabad (AHM), Mohali (MHL), Gadanki (GDN), and Sonipat (SNT). International stations include Mauna Loa (MLO), South Carolina (SCT), Shenandoah National Park (SNP), Walnut Grove, (WGC), Moody (WKT) and Boulder (BAO). For all international stations except BAO, the five-year average (2018 - 2022) has been chosen for the seasonality. For BAO, 2011 – 2016 has been used. The monthly average of the entire study period (February 2023 – January 2025) has been used for this comparison.

Sonipat exhibits very high seasonal amplitude (~ 60 ppm) compared to other sites (~ 15 ppm) across the globe, however, the seasonal amplitude is around 35 ppm at Ahmedabad.





The major attribution to the high seasonal amplitude of $CO_2$ at Sonipat occurs during post-monsoon (Fig. 6a). This high amplitude for November has been observed during 2023 and 2024 (see Fig. 2), a characteristic of the Sonipat station. The high seasonal amplitude is associated with the crop residue burning season over Haryana and Punjab (further discussed in section 3.5). Being surrounded by agricultural land, Sonipat is prone to emissions from crop residue burning. The location of the measurement site in IGP on the downwind of Punjab is a major reason for this strong seasonal variability compared to other sites in the same latitudinal band.

**3.3 Diurnal variability**

Figure 7 (a-d) presents the averaged diurnal variation of atmospheric $CO_2$ mole fraction and PBLH at Sonipat during four seasons for the first year of the study (2023). The diurnal variability has been examined separately for the two years to exclude the influence of growth rate on the diurnal amplitude. Figure S4 presents the diurnal variation for the second year of the study. All the seasons exhibit a similar diurnal pattern with maximum mole fraction in the early morning hours (05:00 - 08:00 am) and minimum mole fraction during the late afternoon hours (2:00 - 3:00 pm). Figure 7(e) shows the monthly average variation in diurnal amplitude during the study period. The difference between the maximum and minimum mole fraction of $CO_2$ in the diurnal cycle is defined as the diurnal amplitude. The diurnal amplitude shows large month-to-month variation with an increasing trend from May to September 2023 and a decreasing trend till February 2024. The lowest diurnal amplitude of about 29 ppm is observed in May, while the highest amplitude at about 63 ppm is observed in September/October (Figure 7(e)). We found that the post-monsoon season exhibited the highest diurnal variability (~ 60 ppm), followed by the pre-monsoon season (~ 35 ppm), winter season (~ 30 ppm) and the monsoon season (20 ppm). The same was observed for 2024 as well.



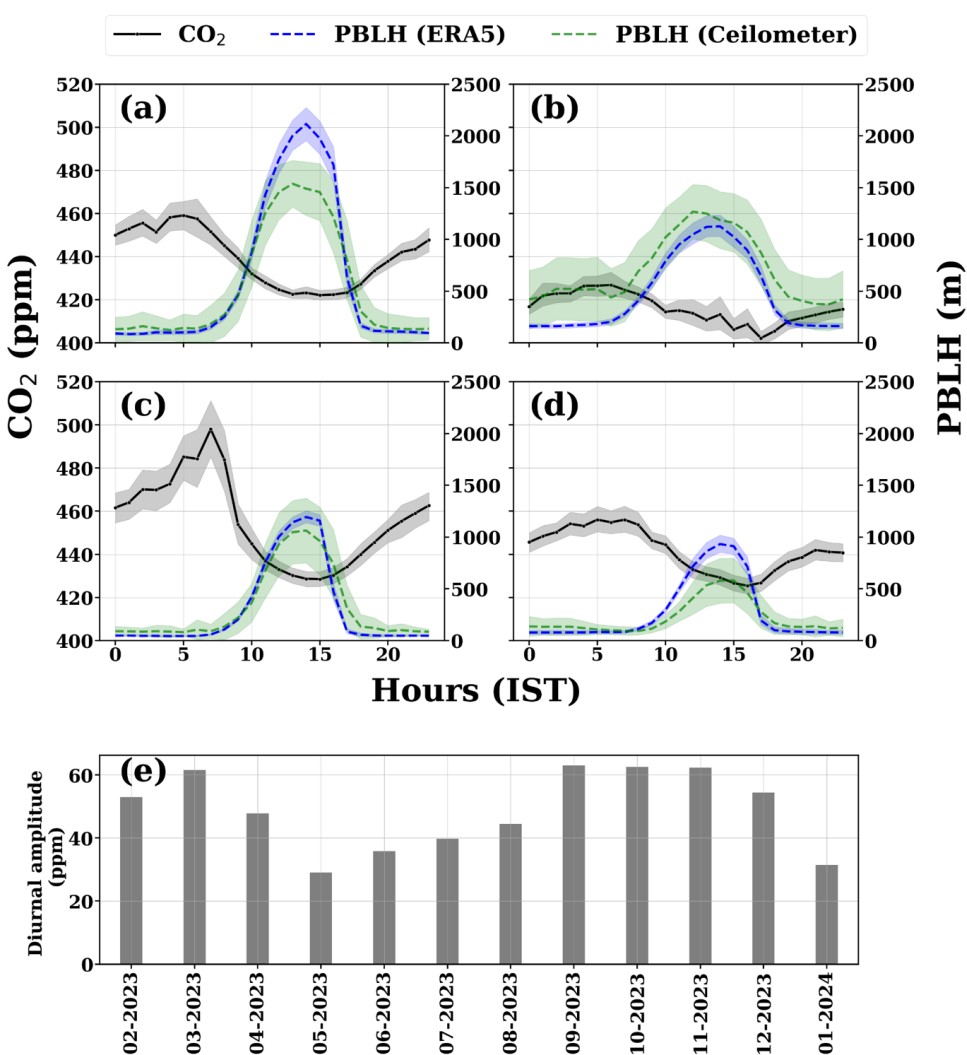

**Figure 7:** (a-d) Seasonally-averaged diurnal variation of atmospheric $CO_2$ over the Sonipat station during the pre-monsoon (MAM), monsoon (JJAS), post-monsoon (ON) and winter (DJF) seasons with planetary boundary layer heights (blue denotes PBLH from ERA5 and green denotes PBLH derived from Ceilometer), (e) monthly variation of the diurnal amplitude of $CO_2$ from February 2023 to January 2024.

The seasonal differences observed in the $CO_2$ diurnal amplitudes can be attributed to changes in local meteorology and biospheric activity over the monitoring station. A key factor that drives the diurnal variability of $CO_2$ is the PBLH, which depends on local meteorology.





The observed diurnal cycle of $CO_2$ is closely associated with the diurnal variation of the PBLH
(Fig. 7). Figure S5 presents the seasonal variation of $CO_2$ compared with PBLH derived from
Ceilometer and ERA5 reanalysis data. It is observed that the $CO_2$ mole fraction steadily
increases throughout the night, reaching its peak in the early morning hours. This accumulation
of $CO_2$ during the night-time can be attributed to poor mixing conditions due to shallow PBLH
and biospheric respiration (Reid and Steyn, 1997). Similarly, minimum mole fraction are
visible during the late afternoon hours (2:00 - 3:00 pm), irrespective of season, when the
boundary layer is well mixed. The peak in $CO_2$ mole fraction during the morning hours can be
attributed to the fumigation effect, a significant rise in surface pollutant mole fraction notable
during the early morning hours due to the breakdown of the nocturnal inversion layer following
sunrise (Stull, 1988). The fumigation effect is pronounced in the post-monsoon and winter
seasons due to weak winds and a shallow PBLH.

Another key driver of $CO_2$ diurnal variability at Sonipat is the photosynthetic activity
of the surrounding vegetation, a characteristic observed in rural areas with vegetative cover
(Imasu & Tanabe, 2018). The combined effect of photosynthetic activity and a well-mixed
PBLH (~1.3 km) during the afternoon hours in the post-monsoon season results in high diurnal
variability. The low mole fraction during afternoon hours in the pre-monsoon season can be
attributed to the dense PBLH (~ 2 km). The low diurnal variability during winter is due to the
shallow PBLH (~ 900 m) and local meteorological conditions like weak winds. The delay in
the evolution of the boundary layer could potentially result in a delayed, more pronounced
fumigation peak during the winter season. Vegetative uptake of $CO_2$ is maximum during the
monsoon season, and the poorly mixed PBL (~ 1.1 km) due to cloudy conditions contributes
to minimum mole fraction and variability of $CO_2$ during this season. The diurnal variation of
GHGs reported by several studies (Nishanth et al., 2014; Patil et al., 2014; Sharma et al., 2014)
from different parts of the country shows a similar trend.
**3.4 Drivers of $CO_2$ variability**

The contribution of biospheric fluxes in driving the $CO_2$ mole fraction over Sonipat (for
2023) was analysed in Fig. 8. Figure 8(a) shows the simulated data from the Mi CASA
biosphere model along with monthly averaged mole fraction of $CO_2$ and daytime $CO_2$ (06:00
– 18:00). The NEE flux represents the net carbon exchange between terrestrial ecosystems
(difference between Rh and NPP). NPP is the net amount of $CO_2$ retained in the biosphere. Rh
is the amount of $CO_2$ emitted into the atmosphere due to the decomposition of organic matter




by microorganisms in the soil. Figure 8(b) presents the simulated NEE from GEOS-Chem and GPP from FluxSat along with monthly averaged mole fraction of daily-mean and nighttime $CO_2$ (18:00 – 06:00). The GPP fluxes, a measure of carbon uptake by plants is also very high during this time. Reco the sum of Ra (autotrophic respiration) and Rh has been calculated as the difference of FluxSat GPP and GEOS-Chem NEE. Positive NEE values suggest an exchange of $CO_2$ from the biosphere to the atmosphere. On the other hand, a negative NEE value (when NPP exceeds Rh) suggests the uptake of $CO_2$ from the atmosphere to the biosphere.

The NEE flux presents a strong positive in June, followed by a gradual decrease up to October (monsoon). During this time, Reco, Rh and GPP exhibit strong enhancements. These enhancements are accompanied by the drawdown of $CO_2$ during this time. The driving factor for this drawdown of $CO_2$ during monsoon is the enhanced ecosystem productivity during this time. Interestingly, post-monsoon and winter months exhibit weak or negative NEE. This is because the Rh values are low during these seasons due to the drier soil conditions and the lack of soil moisture. It is also notable that GPP is very low during these months, which is associated with high $CO_2$ mole fraction as well. This increase in mole fraction is not only due to the lack of vegetation but also due to contributions from other local sources as well.

Figure 8(c) presents the correlation heatmap of all the variables. GPP, Rh and Reco exhibit an inverse correlation with $CO_2$. A strong inverse correlation of $CO_2$ with GPP suggests that the primary sink of $CO_2$ over Sonipat is biospheric activity. It is notable that GPP exhibits a strong positive correlation with Rh and Reco. This is due to the abundance of vegetation from the enhanced soil moisture during monsoon, suggesting that biospheric activity plays a key role in driving the $CO_2$ dynamics over Sonipat.



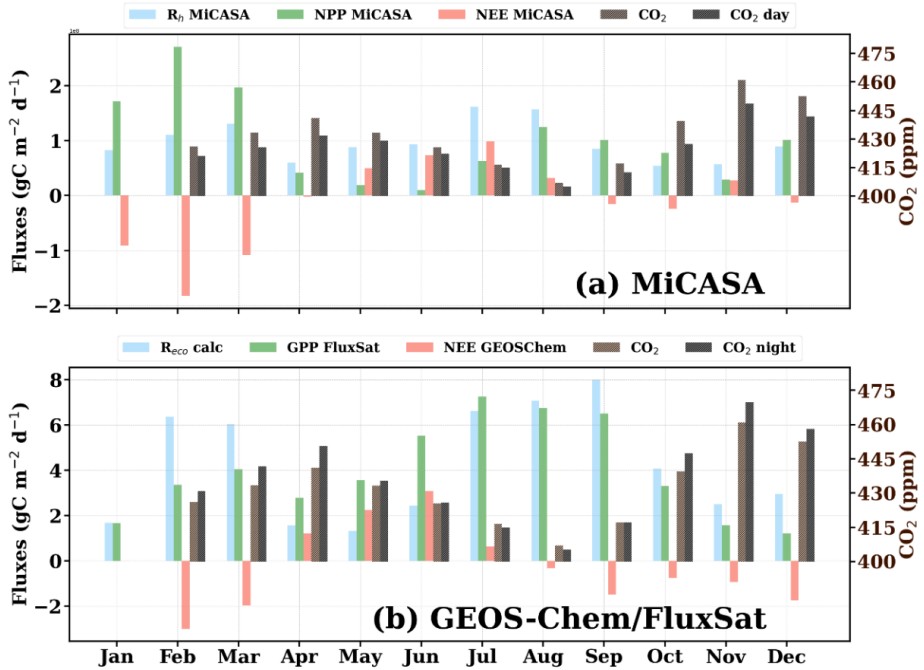

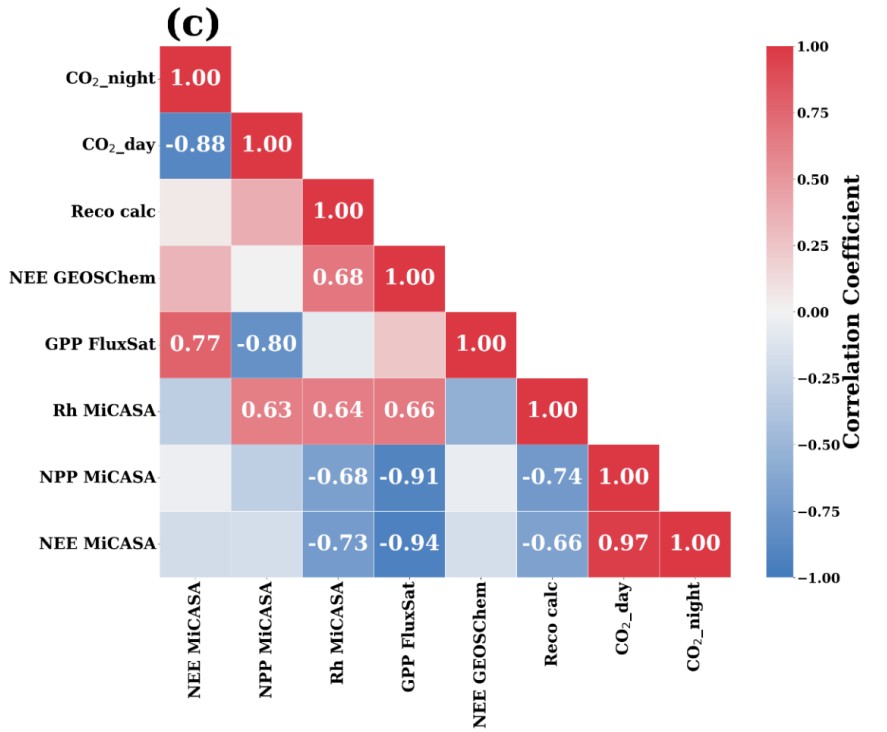





**Figure 8:** Monthly variation of atmospheric $CO_2$ mole fraction (for 2023) over the Sonipat monitoring station compared against (a) biospheric fluxes from the MiCASA terrestrial biospheric model and (b) GEOS-Chem model and FluxSat GPP data. (a - b) The $CO_2$ mole fraction are daytime-mean (06:00 - 18:00 LT) and night time-mean (18:00 - 06:00 LT). The correlation heatmap of all the variables. The annual growth rate of $CO_2$ has been subtracted from the $CO_2$ mole fraction using background data from the Mauna Lou observatory. The variable "Reco calc" was calculated as the difference between NEE (GEOS-Chem) and GPP (FluxSat). The Pearson correlation coefficients with a p value less than 0.05 have been displayed in the correlation plot.

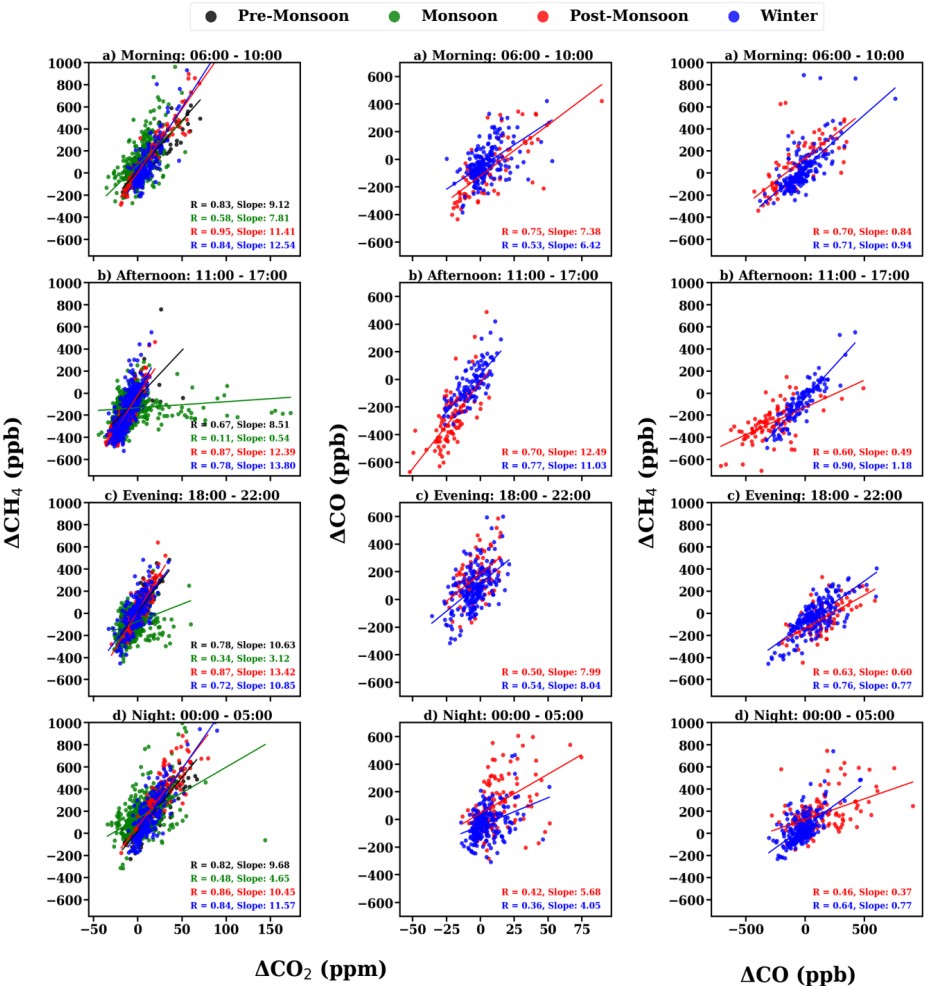





**Figure 9:** Tracer to tracer relations of $\Delta CO_2 / \Delta CH_4$ (left panel), $\Delta CO_2 / \Delta CO$ (middle panel)
and $\Delta CH_4 / \Delta CO$ (right panel) during a) Morning (0600–1000 IST), b) afternoon (1100–1700
IST), c) evening (1800–2200 IST) and d) night (0000–0500 IST).

**3.5 Tracer-Tracer relationships**
The ratios (tracer-tracer) of GHGs have been widely used in previous studies to estimate
different emission source contributions to atmospheric GHGs (Chandra et al., 2016, 2019; Lin
et al., 2015; Lopez, 2012; Paris et al., 2008; Sreenivas et al., 2016, 2022). We follow a similar
tracer-tracer correlation analysis though the stations are in different geographical locations,
however this will help to assess the synoptic variation of $CO_2$ at different diurnal time windows
to understand the emission sources contributing to $CO_2$ mole fraction over Sonipat (Fig. 9).
The measurements have been divided into four-time windows: (a) morning hours (06:00 -
10:00; the PBLH starts to develop after sunrise; local traffic is high), (b) afternoon hours (11:00
- 17:00; the PBLH is well-developed; relatively minimum local traffic, (c) evening hours
(18:00 - 22:00; rush hour traffic and high household emissions), and (d) night hours (00:00 -
00:05; relatively less anthropogenic emission sources). Excessive mole fraction were used in
the correlation analysis to remove the influence of background mole fraction on the correlation
ratios (Worthy et al., 2009). The correlation between the different gases ($CO_2$, $CH_4$, and CO)
has been studied using the robust linear fit regression method.

**3.5.1 Correlation between $CO_2$ and $CH_4$**
Figure 9 (left panel) presents the correlation of excess mole fraction of $CH_4$ and $CO_2$ during
the four seasons. The $CH_4/CO_2$ correlation reveals a strong correlation ($r > 0.6$) for all seasons
except monsoon during all time windows, which suggests a similar source mechanism or a
controlling emission process for both gases at the measurement site. Around the study location,
vehicular emissions from the nearby highway and natural gas combustion emissions are
possible sources. Also, a positive correlation suggests the dominance of anthropogenic
emissions on the carbon cycle over Sonipat (Fang et al., 2015). During monsoon season, the
afternoon time window has a weak correlation with other time windows, revealing the different
source and sink mechanisms of $CO_2$ and $CH_4$, such as the loss of $CH_4$ by hydroxyl radical and
the uptake of $CO_2$ by plants. The regression slope exhibits a higher slope during the post-
monsoon and winter months, and this is associated with the lack of photosynthetic activity and
the dominance of local emissions and long-range transport. The lower values during pre-
monsoon and monsoon seasons are associated with the dominance of vegetation and





photosynthetic activity (terrestrial uptake of $CO_2$). The regression slope shows strong diurnal
variation throughout all seasons. Similar studies across India have presented similar results
with high regression slopes during post-monsoon and winter in comparison to pre-monsoon
and monsoon seasons (Lin et al., 2015; Sreenivas et al., 2016, 2022).

**3.5.2 Correlation between $CO_2$ and CO**
Figure 9 (middle panel) presents the correlation of excess mole fraction of CO and $CO_2$ during
post-monsoon and winter seasons. The $CO/CO_2$ correlation shows strong diurnal variability
suggesting the dominance of different source mechanisms throughout the day, with strong
correlation during the morning and afternoon hours (suggesting a similar source) and a weaker
correlation during the evening and night hours (suggesting different sources) between these
gases. The postmonsoon season has higher regression slopes, which can be associated with the
lack of photosynthetic activity combined with the crop residue burning. The $CO/CO_2$ ratio over
Sonipat ($4 - 12.5$ ppb ppm$^{-1}$) is lower than those for fresh plumes from wildfire (Andreae and
Merlet, 2001; Mauzerall et al., 1998) and much lower than that from biomass burning events
(Matsueda et al., 1999). The low ratios of $CO/CO_2$ can also be due to the contribution of biofuel
burning (which has a higher burning efficiency) during post-monsoon and winter (Andreae and
Merlet, 2001). Lin et al. (2015) reported $CO/CO_2$ ratios of 13 ppb ppm$^{-1}$ over Southeast Asian
outflow from February to April 2001. This value was suggested to be not only due to
biomass/biofuel burning but also due to fossil fuel emissions (Russo et al., 2003), crop residue
burning, and biofuel burning have a combined influence on $CO_2$ and CO mole fraction over
Sonipat during post monsoon and winter. Although there is a contribution of CO and $CO_2$ from
long-range airmass transport (influence of crop residue burning over Punjab) from the
northwest side of the monitoring station, the effect is diluted by other sources. Figure S2
presents the wind patterns during the different seasons, revealing the predominant winds from
the northwest during the post-monsoon season.

**3.5.3 Correlation between $CH_4$ and CO**
Figure 9 (right panel) presents the correlation of excess mole fraction of $CH_4$ and CO during
post-monsoon and winter seasons. The $CH_4/CO$ correlation reveals a stronger correlation ($r >$
0.7) during winter compared to post-monsoon during all time windows, which suggests similar
sources during winter, and a relatively weaker correlation in post-monsoon reveals different
sources. The regression slope exhibits a higher slope during winter compared to post-monsoon.
This is associated with the lack of photosynthetic activity and the dominance of local emissions





and long-range transport. The $CH_4/CO$ ratios range from 0.3 to 1.2 over Sonipat, indicative of
anthropogenic emission sources (Bakwin et al., 1995; Harriss et al., 1994; Lai et al., 2010; Lin
et al., 2015; Niwa et al., 2012; Sawa et al., 2004; Wada et al., 2011; Xiao et al., 2004). The
$CH_4/CO$ ratios range of 0.07 – 0.3 indicates the contribution from biomass and biofuel burning
(Andreae and Merlet, 2001; Mauzerall et al., 1998; Mühle et al., 2002).

Lin et al. (2015) presented similar ratios of $CH_4/CO$ over Pondicherry (PON) and Port

Blair (PBL). High $CH_4$ emissions from livestock can raise the low $CH_4/CO$ ratios from biomass
burning. $CH_4$ and CO emissions from biomass, biofuel burning and livestock estimated from
EDGAR v4.2, 2011 indicate a $CH_4/CO$ ratio of 0.64 – 0.69 over the Indian subcontinent from
2000-2008. These ratios are comparable to the ratios observed during both seasons over
Sonipat.

**4. Conclusions**
This study investigated the high temporal variability of atmospheric $CO_2$ mole fraction at
Sonipat, a suburban station in the Indo-Gangetic Plain. Sonipat's location, being in the
downwind of Punjab and upwind of Delhi, makes it an ideal site for examining the influence
of different regional air masses in the IGP region. The atmospheric $CO_2$ mole fraction from
February 2023 to January 2025 have been measured with a GHG analyser using the laser-based
cavity ring-down spectroscopy technique. To understand the key drivers of seasonal and
diurnal $CO_2$ variability over the Sonipat station and the IGP region, we used a combination of
ground-based and satellite-based measurements, three different model outputs, ecosystem
proxy variables, and tracer-tracer analysis technique.
The salient findings from this study are listed below.
●   The surface-based measurements of atmospheric $CO_2$ mole fraction exhibit the large $CO_2$

seasonality with maximum mole fraction (456.4 ppm) during post-monsoon and minimum

mole fraction (407.2 ppm) during monsoon, with an average mole fraction of 422.6 ppm.

●   The comparison of the seasonality of atmospheric $CO_2$ over Sonipat with other Indian and

global sites in the same latitudinal band reveals very high seasonality over that is observed

at Sonipat. This high seasonality is attributed to the high mole fraction of $CO_2$ during

November (post-monsoon) from local emissions and crop residue burning. The location

of the measurement site in the IGP region on the downwind of Punjab is a major reason

for this strong seasonal variability compared to other sites in the same latitudinal band.



- Our results also indicate that the biospheric activity was the primary driver of $CO_2$ seasonal variability over Sonipat, with anthropogenic emissions and soil respiration as the major sources and photosynthetic carbon uptake as the major sink. In addition, the boundary layer dynamics and air mass transport from upwind regions significantly contribute to the build-up of $CO_2$ mole fraction.

- Although both the CarbonTracker and MIROC-ACTM models could capture the broad seasonal pattern of $CO_2$ mole fraction, the models substantially underestimated the $CO_2$ mole fraction. The OCO-2 and OCO-3 satellite $XCO_2$ retrievals also revealed similar seasonal variability; however, the satellites could not capture $CO_2$ enhancements due to local sources.

- The atmospheric $CO_2$ mole fraction at Sonipat exhibit a consistent diurnal pattern irrespective of season, with an observed maximum during morning hours, which can be attributed to the fumigation effect, with a gradual decrease during the day and a minimum during afternoon hours with enhanced photosynthetic activity. A slight shift in time for the morning peaks was observed from season to season due to the change in the time of sunrise, resulting in a shift in photosynthetic activity. The diurnal amplitude of $CO_2$ was observed to peak in post-monsoon (maximum in November) and draw down (minimum in May) in monsoon.

- The tracer-tracer relationships during different time periods for the post-monsoon and winter seasons were examined. Analysis reveals that $CO_2$ and $CH_4$ show a strong positive correlation during all seasons, and the higher slopes are due to the lack of photosynthetic activity and the influence of local winds. This strong correlation suggests common anthropogenic sources for both these gases. The $CO/CO_2$ ratios reveal the influence of long-range transport of crop residue burning over Punjab on $CO_2$ mole fraction in Sonipat during post-monsoon.



**Data availability**

- The OCO-2 and OCO-3 data is downloaded from https://disc.gsfc.nasa.gov/datasets/. This study utilizes the bias-corrected OCO-2 v11.1r data product (https://disc.gsfc.nasa.gov/datasets/OCO2_L2_Lite_FP_11.1r/summary?keywords=oco2) and the OCO-3 v10.4r data product (https://disc.gsfc.nasa.gov/datasets/OCO3_L2_Lite_FP_10.4r/summary?keywords=oco3).

- The CT-2020 model outputs were downloaded from https://gml.noaa.gov/aftp/products/carbontracker/co2/. The CASA model outputs were downloaded from https://disc.gsfc.nasa.gov/datasets/GEOS_CASAGFED_M_FLUX_3/summary?keywords=CASA.

- The ERA5 reanalysis datasets were downloaded from https://cds.climate.copernicus.eu/cdsapp#!/dataset/reanalysis-era5-single-levels.

- The satellite estimates of NDVI were downloaded from https://www.ncei.noaa.gov/data/land-normalized-difference-vegetation-index/access/.

- This study utilises bias-corrected SIF data from OCO-2 v11r data product (https://disc.gsfc.nasa.gov/datasets/OCO2_L2_Lite_SIF_11r/summary?keywords=oco2%20sif).

- The FluxSat data is downloaded from https://avdc.gsfc.nasa.gov/pub/tmp/FluxSat_GPP/. This study uses FluxSat version 2.2 dataproduct.

- The ObsPack data is available at https://gml.noaa.gov/ccgg/obspack/data.php. This study used ObsPack V2.0 dataproduct.

**Acknowledgements:**

We acknowledge institutional support and funding provided by IIT Delhi and other stakeholders to develop the IIT Delhi Atmospheric Observatory at Sonipat. In particular, we thank Shahzad Gani (IIT Delhi) for his contribution to the observatory. We thank the Aakash Project team for providing trace gas data from the CUPI-G sensors. We acknowledge the OCO-2, OCO-3, CASA, CarbonTracker, and ERA5 teams for providing the data used in this study.



**Author Contributions:**
**Conceptualization:** VJV, RKK, SP
**Data curation:** VJV, RKK, JR, DG, SD, YM, PKP
**Investigation, Methodology:** VJV, RKK, SP, PKP
**Software, Visualisation:** VJV
**Writing – original draft:** VJV
**Writing – review & editing:** RKK, JR, DG, SD, YM, PKP

**Competing interests**
The authors declare that they have no conflict of interest.



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
