# Peer review of "Insights into the high temporal variability of atmospheric carbon dioxide"

_EGUsphere, 2025_

## Referee Comment (RC2)

The authors present a comprehensive data analysis of CO2 observations from an India monitoring station, which is a relevant contribution to this special issue about GHG in the Asia-Pacific region. The analysis, which includes related parameters like PBLH, NEE, CH4, CO, is also competently discussed.

However, the work is primarily descriptive, presenting a common data analysis without showing substantial scientific depth for the scope of ACP. The manuscript's emphasis on reporting and discussing measurements appears more aligned with the scope of AMT or a measurement report.

**Major comments:**

Line 588: emissions from vehicular and natural gas combustion may exhibit different patterns. Could the authors elaborate on this? Additionally, as mentioned before, biomass burning is also a potential source -- how is this reflected in the correlation data? This is particularly interesting given Sonipat's proximity to Delhi. Could the authors clarify how urban emissions from the city contribute to the observed enhancements? Furthermore, the correlation analysis could be enhanced by incorporating wind direction.

How does the satellite-derived tracer-tracer correlations?

Line 594-598: The description of deltaCH4/deltaCO2 appears contradictory. During periods of low photosynthetic activity (post-monsoon and winter), CO2 enhancement should increase, resulting in lower slope values. Conversely, higher photosynthetic activity during the monsoon season should lead to elevated slope values. However, the data show that deltaCO2 levels are significantly higher during monsoon afternoons compared to other periods. Could the authors explain this inconsistency?

Line 596: what does the "long-range transport" indicate?

Line 603: what are the correlations between deltaCO and deltaCO2 during the pre-monsoon and monsoon seasons?

Line 646-648: The authors highlight that Sonipat's location—between Punjab and Delhi, with predominant northwesterly winds—makes it an ideal site for studying the influence of different regional air masses in the IGP region. However, this potential is not fully explored in the manuscript, aside from brief mentions in the tracer-tracer correlations section. Could the authors expand on this aspect?

**Technical comments:**

Line 25: better specified the month period for "monsoon" and "post-monsoon"

Line 70: column >>> columnar

Line 107: two "such as"

Line 136: >>> background of CO2…

Line 252: a citation is missing

Line 283: Figure 1(a) better add year on the x-axis

Line 311: what's the sources on the northwest of study site?

Line 335: Figure2(a) shows minimum values in PBLH in January and highest values in April/May. The monsoon period covers June to September. Doesn't mean that the maximum values also occur during pre-monsoon months? Please explain this.

Line 344: Figure 2(b) NDVI shows an increase during November – April, even surpassing its summer values. However, this pattern is not reflected in the SIF data. Could you please explain this?

Line 434: Figure 6: Sonipat monitoring station is presented as "SNPT", which is "SNT" in the text. Please modify.